# Material Data Identification in an Induction Hardening Test Rig with Physics-Informed Neural Networks

**DOI:** 10.3390/ma16145013

**Published:** 2023-07-15

**Authors:** Mohammad Zhian Asadzadeh, Klaus Roppert, Peter Raninger

**Affiliations:** 1Materials Center Leoben Forschung GmbH (MCL), Roseggerstraße 12, 8700 Leoben, Austria; 2Institute of Fundamentals and Theory of Electrical Engineering, Technical University of Graz, Inffeldgasse 18/I, 8010 Graz, Austria

**Keywords:** neural networks, inverse problems, PINNS, induction heating, material data

## Abstract

Physics-Informed neural networks (PINNs) have demonstrated remarkable performance in solving partial differential equations (PDEs) by incorporating the governing PDEs into the network’s loss function during optimization. PINNs have been successfully applied to diverse inverse and forward problems. This study investigates the feasibility of using PINNs for material data identification in an induction hardening test rig. By utilizing temperature sensor data and imposing the heat equation with initial and boundary conditions, thermo-physical material properties, such as specific heat, thermal conductivity, and the heat convection coefficient, were estimated. To validate the effectiveness of the PINNs in material data estimation, benchmark data generated by a finite element model (FEM) of an air-cooled cylindrical sample were used. The accurate identification of the material data using only a limited number of virtual temperature sensor data points was demonstrated. The influence of the sensor positions and measurement noise on the uncertainty of the estimated parameters was examined. The study confirms the robustness and accuracy of this approach in the presence of measurement noise, albeit with lower efficiency, thereby requiring more time to converge. Lastly, the applicability of the presented approach to real measurement data obtained from an air-cooled cylindrical sample heated in an induction heating test rig was discussed. This research contributes to the accurate offline estimation of material data and has implications for optimizing induction heat treatments.

## 1. Introduction

Inverse problems are interesting in science and engineering, because they provide insights into the unknown characteristics of a system. Sensor data is often required, along with the governing PDEs/equations, to directly or indirectly solve for the target latent variables or identify the system’s characteristics. Heat transfer and thermal problem identification are involved in many industrial applications, such as the induction hardening process. Accurately identifying thermal problems requires knowledge of the heat flows at the boundaries, the initial and boundary conditions, and precise thermo-physical material data.

Traditionally, trial and error, along with sophisticated and expensive approaches, have been applied to obtain such critical knowledge about the system, when they have been feasible in practice. More rigorous mathematical approaches have also been developed and applied to solve inverse heat conduction problems, such as the regularization approach of Tikhonov [1], linear least square methods [2], and genetic algorithm-based solutions [3]. Depending on the settings of the heat conduction problem, such as the type of boundary conditions and the number of unknown parameters, the commonly existing approaches might not fully or accurately identify the inverse problem.

Furthermore, inverse problems are often ill-posed due to measurement errors or the incompleteness of temperature sensor information. Therefore, the approaches of developing, investigating, and exploiting new approaches to tackle more complex inverse problems are interesting from both a research and application standpoint.

Artificial neural networks, since their original single feed-forward perceptron version introduction in 1958 by Frank Rosenblatt, have contributed to a diversity of computational tasks, such as image processing [4], computer vision [5], autonomous driving [6], e-commerce [7], education [8], voice assistance [9], spam filters [10], and more. Due to advances in hardware and software technologies, machine learning approaches have been exploited in various scientific and engineering problems, such as fluid mechanics [11], earthquake prediction [12], milling processes [13], to name some examples.

Recently, a new variant of neural networks known as physics-informed neural networks (PINNs) has attracted a lot of attention since their introduction by Raissi et al. in [14]. Combining the measurement data and governing PDEs during the training of PINNs has enabled the solution and inversion of the underlying equations of a diversity of physical systems [14,15,16,17]. PINN models are more interpretable, have higher extrapolation power, and require less data for their training compared to their standalone neural network counterparts. Moreover, PINNs are an attractive alternative and might have advantages over FEM solutions where the boundary conditions are not known or the information about the physical system is not complete. The authors in [18] have provided a review of the current trends in integrating physics into machine learning. They discussed the state-of-the-art capabilities and limitations of PINNs in solving forward and inverse problems, as well as their potential to discover embedded physics in high-dimensional problems.

Neural-Network-Based approaches have been applied to solve forward and inverse heat problems. In [19], the authors applied an artificial neural network to estimate the thermal conductivity of an orthotropic medium. A few recent works have also applied PINNs to solve heat problems. The authors in [20] applied PINNs to a 1D unsteady heat conduction problem with Neumann boundary conditions on both sides. They estimated the time-dependent heat flux on the boundary alongside the heat diffusivity using provided temperature information at certain locations in the solution domain. In an extensive work reported in Ref. [21], PINNs were applied to various heat transfer problems. In the forced and mixed convection problem with unknown boundary conditions on the heated surfaces, the temperature and velocity fields were obtained in the domain using very sparse temperature measurements. The Stefan problem for a two-phase flow was also identified in 1D given a few temperature measurements in the domain. The forward solution of the 1D-2D heat transfer problem was obtained using PINNs and was found to be valid for different convection boundary conditions [22]. In [23], the forward and inverse 1D heat conduction problem with Dirichlet boundary conditions was solved to identify constant and space–time varying unknown parameters.A PINN was also developed to solve a forward transient heat problem in 1D and 2D with convective boundary conditions that was applicable to heated parts in industrial ovens [22]. A physics-informed Bayesian neural network was applied to solve forward and inverse heat conduction problems considering different types of boundary conditions for a 2D plate and 3D heat sink [24]. In the inverse problem, the boundary conditions and the diffusion coefficient were identified with their estimated uncertainties. In [23], a new PINN architecture was proposed in which coupled neural networks were used to estimate temperature-dependent thermal conductivity in an inverse 1D transient heat problem. In [25], A PINN was applied to solve steady and transient heat conduction problems that was applicable to functionally graded materials, but only the radial and temporal coordinates were considered. In [26], a PINN was employed with a Runge–Kutta discrete-time scheme to solve the transient heat transfer problem of three-dimensional functionally graded materials.

The current works have mostly focused on solving forward and inverse heat transfer problems, which are often limited to low dimensions, specific boundary conditions, and restricted to rectangular geometries. The aim of this study is to address the forward and inverse 2D transient heat transfer problem in cylindrical bars that is relevant to the induction hardening process. The focus is on the air cooling of cylindrical samples with the consideration of realistic boundary conditions, such as convection and radiation governed by Neumann boundary conditions. The temperature evolution in this process relies on crucial thermo-physical material data, including mass density, specific heat, thermal conductivity, as well as heat convection and emissivity coefficients.

The primary objective of this research is to investigate the applicability of physics-informed neural networks (PINNs) in identifying material and process data in the context of a 2D transient heat transfer problem. The goal is to assess the accuracy of identifying these data using only a limited number of temperature sensor measurements within the cylindrical samples. By incorporating the governing physics of the cooling process as soft constraints within the objective function, the PINN network is optimized to solve both the forward and inverse problems.

The outcomes of this study have significant relevance for the field of material science and engineering. The successful identification of material data and boundary conditions with minimal uncertainty provides valuable insights into the behavior and properties of materials that are utilized in the induction hardening process. This knowledge can contribute to optimizing heat treatment procedures, thereby resulting in the enhanced performance of heat-treated materials. Furthermore, the methodology developed in this study holds potential for broader application to other material systems and engineering processes, thereby offering a promising data-driven approach for analysis and optimization across various industrial applications.

Finally, the study contributes to the advancement of the field by demonstrating the effectiveness of PINNs in material data identification. PINNs offer a promising approach for solving complex inverse problems in various scientific and engineering domains. The ability to accurately estimate material properties using limited sensor data showcases the potential of PINNs as a valuable tool in data-driven engineering applications.

The manuscript is organized as follows. Section 2 provides a brief overview of the induction hardening process steps. In Section 3, the governing equations and the Finite Element Method (FEM) model are presented, which are utilized to generate the benchmark data for free air cooling. Section 5 introduces the architecture of the PINN model, along with the constraints applied. The results of solving the forward and inverse problems of the air-cooling step in the induction hardening process are presented in Section 6. Finally, Section 7 summarizes the findings and provides concluding remarks on the work.

## 2. Induction Hardening Process

In the metal production and processing industry, induction hardening is a widely used process that is known for its effectiveness in metal hardening. This process relies on electromagnetic induction, where a metallic object is heated by resisting the induced eddy currents. It encompasses several key steps, including austenitization, quenching, tempering, and air cooling. The duration and process parameters of each step depend on the desired outcome of the hardening process. To model the entire process accurately, a comprehensive multi-physics simulation is required, which addresses thermal, electromagnetic, metallurgical, and mechanical aspects. For a detailed understanding, we refer readers to Ref. [27] and related references.

To achieve precise modeling, it is essential to have thorough knowledge of the material data, boundary conditions, and magnetic properties involved. In the case of air cooling, crucial material parameters such as thermal conductivity, mass density, and specific heat come into play. Additionally, the boundary conditions incorporate factors such as the heat convection coefficient and emissivity, which are primarily influenced by the environmental setup of the process (refer to Equation (Equation 1)). In this study, our focus is on the inverse solution of the air-cooling process, which aims to identify the material data and boundary conditions using a limited number of temperature sensor measurements. We specifically chose the cooling process step, because it does not involve the coupling of thermal and electromagnetic problems (as explained in Section 3), yet it offers valuable access to crucial thermo-physical material data.

## 3. Finite Element Model and Benchmark Data

Finite element modeling (FEM) was employed to simulate the air cooling process and to generate benchmark data for the feasibility study of material parameter identification using PINNs. The FEM solution was obtained using the open source software OpenCFS [28]. The governing partial differential equation (PDE) is a heat equation without a volumetric heat source term, and it is accompanied by initial and boundary conditions (ICs and BCs) as follows:(1)$$\begin{array}{cc} PDE::\rho_{m}C_{p}\frac{\partial T}{\partial t} & =\vec{\nabla }\cdot \left(\lambda \vec{\nabla }T\right)\\ IC::T(r,z,t=0) & =T_{init}\\ BC::-\lambda \partial T/\partial \hat{n} & =h\left(T-T_{amb}\right)+\sigma_{B}\epsilon \left(T^{4}-T_{amb}^{4}\right). \end{array}$$

In the governing equation, the symbols represent the following physical quantities: $\rho_{m}$ denotes the mass density, $C_{p}$ represents the specific heat, λ represents the thermal conductivity of the material, *h* denotes the heat convection coefficient, ϵ represents the emissivity coefficient of the material, $T_{init}$ represents the initial temperature of the sample, $T_{amb}$ denotes the ambient temperature, and $\sigma_{B}$ represents the Stefan–Boltzmann constant. The symbol $\hat{n}$ denotes the unit vector normal to the surfaces of the sample, as shown in Figure 1. The sample has a cylindrical geometry, as depicted in Figure 1, and, for the purposes of this study, we consider an axisymmetric problem. Therefore, the temperature depends only on the radial (*r*) and axial (*z*) coordinates, as shown in Figure 1. The cooling process primarily occurs on the boundaries of the sample, specifically in the directions indicated by the surface normal vectors $\hat{r}$, $\hat{z}$, and $-\hat{z}$. In these directions, heat is dissipated through radiation and free convection. The combination of these cooling mechanisms plays a crucial role in regulating the temperature distribution and evolution within the sample during the cooling process.

The FEM simulation data were generated for a sample with a length of L = 0.1 [m] and a radius of R = 0.0125 [m]. The material data and cooling parameters (*h* and ϵ) considered in this work are reported in Table 1 and are fixed, unless otherwise stated. The material data correspond to a non-magnetic sample, such as Aluminum.

The main objective of this study was to investigate the feasibility of estimating material data and boundary condition parameters, which were utilized in the FEM simulations as the input (as reported in Table 1) through the application of physics-informed neural networks (PINNs). The estimation was performed using only a limited amount of temperature sensor data. To accomplish this, virtual temperature sensor data were generated and incorporated as observation constraints during the training of the PINN model.

The ultimate aim is to develop a robust framework that can be applied to various material data and cooling conditions, thereby extending its applicability beyond the specific case studied in this work. By successfully demonstrating the feasibility of using PINNs to estimate material data and boundary conditions, this research can provide a valuable tool for accurate parameter identification and optimization in the field of heat transfer and material science.

## 4. Temperature Sensor Data

### Virtual Temperature Sensor Data

In this study, two configurations of virtual sensors were considered, as illustrated in Figure 2. In the first configuration (referred to as “config:1”), the sensors were positioned radially across the sample. In the second configuration (referred to as “config:2”), the sensors were placed at varying axial locations. This choice was motivated by potential limitations in the practical sensor implementation or temperature measurement instrumentation for the sample.

In Figure 2b,c, the temperature evolution of two sensors located on the surface and the center of the sample in two configurations is illustrated. It is observed that, in config:1, there was not a significant temperature difference between the sensors, while, in config:2, the difference was observable. In Figure 2d, the temperature difference between the sensors at the center and surface of the sample is shown. It is true that, due to the considered high value of thermal conductivity (λ = 237), there was not a significant temperature difference across the sample. However, with the chosen material data from Table 1, it is possible to consider two sets of sensor locations with low and moderate temperature resolutions. Therefore, it is interesting for practical applications to investigate the accuracy of the applied approach for material data estimation regarding both sensor arrangements, which correspond to different temperature distributions across the sample.

## 5. PINN Model

PINN models have been extensively investigated and applied in the literature across various engineering fields (interested readers are referred to the relevant cited references in the introduction). PINNs have demonstrated remarkable success in solving problems where complete knowledge of the physics is often lacking, and only a limited amount of noisy data is available. With PINNs, it is possible to fit observation data while adhering to the underlying governing PDEs. This is made possible by automatic differentiation, which is facilitated by computational tools such as TensorFlow [29], which enable the evaluation of differential operators without discretization errors.

The applied PINN model in this work, along with the imposed constraints relevant to the air cooling of a cylindrical sample, are briefly introduced. Figure 3 illustrates the structure of the PINN model, which incorporates the boundary and initial conditions. The PDE given by Equation (Equation 1) is solved using a neural network NN(r,z,t,θ) to approximate the temperature distribution T(r,z,t). The network NN takes the input feature vector (r,z,t) and processes it through fully connected layers with non-linear activation functions f(.) to estimate the target latent function *T*. The network NN consists of trainable parameters θ, including weights and biases, which are optimized during the minimization of the loss function. The loss function incorporates the constraints, including the PDE, initial and boundary conditions, and the available observation data. These constraints act as a soft regularizer for the neural network. Consequently, the loss function is defined as the sum of these terms as follows:(2)$$L\left(\theta ,P;N\right)=w_{pde}L_{pde}+w_{ic}L_{ic}+w_{bc}L_{bc}+w_{ob}L_{ob}.$$

The loss function consists of several components. $L_{pde}$ represents the loss associated with the PDE constraint, $L_{ic}$ represents the loss associated with the initial condition constraint, $L_{bc}$ represents the loss associated with the boundary condition constraint, and $L_{ob}$ represents the loss associated with the observation data. The weights *w* are used to balance the contributions of each loss term. The total number of residual points N is the sum of the number of points for each loss term, including $N_{pde}$, $N_{ic}$, $N_{bc}$, and $N_{obs}$.

In an inverse problem, the network constraints contain unknown parameters denoted by P. However, the available measurement data for the target function, specifically the temperature T(r,z,t), are limited in quantity. These measurements contribute to the observation loss term $L_{obs}$. To quantify the discrepancies between the predicted and target values, the loss terms are evaluated using the standard $L^{2}$ mean squared error (MSE) function.

Training PINN models can be challenging, and they may encounter difficulties or even fail to converge for simple problems. This is primarily due to the varying convergence rates of the different terms in the loss function. To address this issue, adaptive approaches have been proposed, which leverage concepts such as neural tangent kernels and gradient pathologies [30]. Another approach, introduced in [31], is the use of gradient-enhanced PINNs, referred to as gPINNs, which significantly improve the performance and convergence compared to standard PINNs without adaptive weights. In gPINNs, the gradient of the PDE equation with respect to the input features is incorporated into the overall loss function. In this work, we adopted the approach presented in [31], and, thus, we updated the loss function from Equation (Equation 2) as follows:(3)$$L\left(\theta ,P;N\right)=w_{pde}L_{pde}+w_{ic}L_{ic}+w_{bc}L_{bc}+w_{ob}L_{ob}+\sum_{i=r,z,t}w_{pde}^{i}\nabla_{i}L_{pde},$$
in which $\nabla_{i}$ is the gradient operator with respect to feature *i* (i.e., *r*, *z*, and *t*). As shown in [31], the performance of gPINNs depends on the choice of the weights $w_{pde}^{i}$. Based on their investigation, a value around 0.01 was considered appropriate for most problems. The optimal weight value can be determined through a grid search. For our specific application problem, a weight value of 0.01 was found to work well for the considered heat equation. Thus, all the weights were fixed as indicated in Equation (Equation 4) unless otherwise stated.
(4)$$\left[w_{pde},w_{ic},w_{bc},w_{ob},w_{pde}^{i}\right]=\left[1.0,1.0,1.0,10000.0,0.01\right].$$

Imbalanced gradients can arise due to differences in the units of measurement of the loss terms or variations in the number of samples used to calculate them. In the considered inverse transient 2D heat problem, there were only a few samples of observation sensor data available (5 × 200 = 1000), whereas the physical domain was sampled using 16,500 collocation points. To balance the magnitudes of the loss terms, a larger weight ($w_{ob}$ = 10,000) was applied to the loss on the observation data. Although this weight may not be specifically optimized for this problem, it yielded satisfactory results, as the estimated parameters were in good agreement with the true values (as shown in the Results section). It was conjectured that the large weight assigned to the observation data would cause the network to prioritize fitting the available sensor data during the initial stages of optimization. Subsequently, the network was presumed to extend the solution from the measurement points to the rest of the domain by minimizing the other loss terms. This approach leads to a more accurate estimation of the temperature distribution in the sample and identified material parameters. It should be noted that the weight $w_{ob}$ = 10,000 was only applied in the inverse problem for material data identification.

The neural network (NN) used in this study has a moderate size, with a depth of 4 and a width of 10. The hidden layers are equipped with a hyperbolic tangent activation function, while the output layer employs a linear activation function. The Adam optimizer with a learning rate of 0.001 was employed for the optimization process. These hyper-parameters were determined through trial and error and were found to be sufficient for the considered problem.

Hyper-Parameter tuning is a crucial step in any machine-learning-based approach. However, tuning hyper-parameters in PINNs can be challenging, because the performance metric used as an objective function is not well-defined. In standard neural network training, labeled measured data are available, and the error function is constructed based on the deviation between the estimated and measured values. Therefore, to achieve better generalization and performance on unseen test data, one must be cautious not to overfit the training data. However, in PINN models, which solve PDEs as imposed constraints, it is not possible to directly monitor the accuracy (training error) and generalization error (test error), since the true solution is not accessible. Moreover, the focus in PINN design is to achieve a significant reduction in the total loss, which comprises different loss terms. Therefore, hyper-parameter tuning in PINNs aims to find an optimal set of parameters that lead to a substantial reduction in the objective value without significantly increasing the training time. Some works have explored the effect of hyper-parameters on the accuracy of PINNs for various PDE benchmarks [32,33]. Our chosen hyper-parameters yielded satisfactory results without a significant increase in the computational time, and some of these hyper-parameters align with the findings of benchmark studies for the heat equation in [32].

The code was implemented using the DeepXDE library [34] with the Tensorflow backend [29]. In the following sections, we present the performance of the gradient-based PINN (gPINN) that was applied to solve both the forward and inverse problems of an air-cooled cylindrical sample.

## 6. Results

### 6.1. Forward Solution of the Heat Equation

The cooling problem of a cylindrical sample requires solving Equation (Equation 1) in both space and time, thus resulting in a 2D transient heat transfer problem due to the axial symmetry of the solution. The effectiveness of various PINN variations in tackling forward or inverse problems can differ depending on the complexity of the governing PDEs and the specific initial and boundary conditions employed [35,36]. Thus, it is advisable to evaluate the quality of the forward solution using the gPINN model before utilizing it to address the intended inverse problem.

Figure 4 presents the quality of the forward solution obtained using the gPINN model compared to the FEM approximation. The forward problem did not involve the use of temperature sensor data. However, it is crucial to sufficiently sample the solution domain to accurately determine the temperature distribution in space and time, thereby ensuring compliance with the imposed initial and boundary conditions. Various sampling strategies have been proposed in the literature [37,38,39,40] that can be incorporated into PINN models for solving PDEs. However, the effectiveness of each sampling strategy needs to be explored, thereby taking into account the problem’s complexity and the geometry of the domain. In a study by Daw et al. [41], it was demonstrated that random sampling of the solution domain during optimization was effective for solving higher-dimensional problems. Therefore, a dynamic random sampling strategy was adopted in this work, where the domain was re-sampled after every 5000 optimization steps. The number of collocation points used was $N_{pde}$ = 15,000, $N_{bc}=1000$, and $N_{ic}=500$ (as shown in Equation (Equation 3)).

In Figure 4a, the PINN solution of the temperature distribution in the r–z plane at a time frame of t=10 [s] is presented. To generate the temperature distribution plot, a denser grid in space (200 × 200) was used after training the model. The FEM solution was not included in the figure, since it was obtained on a sparser computational grid. Figure 4b shows the temperature evolution at two sensor locations in the sample, thus comparing the FEM and PINN solutions. Both sensor locations exhibited high accuracy in capturing the temperature dynamics. The maximum absolute error between the FEM and PINN results was ΔT=0.15[K]. It should be noted that achieving such a solution with the given domain sampling required a total epoch number of at least $5\times 10^{6}$. On the Nvidia Quadro RTX 5000 GPUs, the forward solution took approximately 9 h.

### 6.2. Inverse Problem: Material Data Identification

In the governing PDE and boundary conditions of the considered free air-cooling problem, there are five parameters involved: mass density $\rho_{m}$, specific heat $C_{p}$, thermal conductivity λ, convective heat transfer coefficient *h*, and emissivity ϵ (see Equation (Equation 1)). However, due to the coupling between $C_{p}$ and $\rho_{m}$ in the governing PDE equation, it is only possible to identify one of the parameters. The identification of mass density $\rho_{m}$ can be achieved using simpler standard approaches compared to the identification of specific heat $C_{p}$. Therefore, in this work, the focus was specifically on identifying $C_{p}$.

Out of the remaining four parameters, only three can be identified due to redundancy in the governing equations (see Equation (Equation 1)). Although there are four unknowns, there are only three independent equations. This redundancy arises from the equivalence of the equations describing the heat flow on the upper and lower boundaries due to symmetry. Consequently, it is only feasible to identify two ratios of the unknown parameters that are coupled to the boundary condition equations. In this investigation, the emissivity ϵ was assumed to be already known. While emissivity is a crucial quantity, it can be well approximated and documented for common surface materials [42]. Therefore, the primary objective of this study was to investigate the feasibility of utilizing gPINNs to solve the inverse problem of a cooling process and to accurately estimate the specific heat $C_{p}$, thermal conductivity λ, and convective heat transfer coefficient *h*. The challenge lies in the fact that only a limited number of temperature sensor data are available for parameter estimation. The goal is to develop an effective methodology that can leverage these sparse measurements to provide accurate estimates of these important parameters in the cooling process.

Two configurations of the virtual temperature sensor data were considered in this study, as described in Section 4. It is important to note that the process data used in this work were fixed to specific values, as provided in Table 1. However, the proposed approach is applicable to any material data and boundary condition parameters. The methodology is not limited to the specific values used in this study and can be adapted to different materials and boundary conditions as required.

For solving the inverse problem, the hype-rparameters of the gPINN model were set to the same values as those used for the forward model, as described in Section 6.1. Additionally, the loss function was modified to include an extra term representing the loss of the observation data, as shown in Equation (Equation 3).

The unknown parameters ($C_{p}$, λ, and *h*) were initially assigned random values within the physically feasible range, thereby ensuring that they were positive at the start of the optimization process for the inverse problem. The optimization process may have required more or fewer than $5\times 10^{6}$ epochs to converge, but this longer duration was chosen to evaluate the stability of the obtained solution and to assess the efficiency of the optimization process.

To sample the feature domain (*r*, *z*, and *t*), the random sampling strategy described in Section 6.1 was applied. This approach ensured that the optimization process explored a wide range of points within the domain, thus facilitating the estimation of the unknown parameters.

In Figure 5a, the identified parameters for sensor configuration 2 are reported. The parameters were observed to be identified quite accurately, and the optimization converged much earlier than in the forward problem. This is because the objective function of the inverse PINN was anchored with temperature sensor data (here, they were virtual, not measured, and they were generated using the FEM). The relative errors of the identified parameters $C_{p}$, λ, and *h* were 2.6%, 5.27%, and 7.35%, respectively. To guide the optimization and ensure that the temperature estimate of the model closely matched the sensor data, a weight of $w_{ob}$ = 10,000 was assigned to the cost term of the observation data (derived from virtual sensors).

In Figure 5b, the temperature difference between the predicted and true values (here, they are the virtual sensor data derived from the FEM, which are the sensor locations of config:2) is shown. A very low error was observed, with the maximum occurring on the surface of the sample at approximately ΔT=0.06 [K]. In Figure 5b, the vertical variation of the error at each sensor location corresponded to different time steps. As mentioned earlier, in sensor config:2, the axial location of the sensors (*z*) was changing.

To investigate the possibility of identifying the parameters with sensor configuration 1, the optimization of the inverse problem using gPINNs was performed. In Figure 6, the results for this sensor arrangement are reported. It is observed that, in this case, the material data were also identified, but not as accurately as in the config:2 case. The relative errors of the identified parameters $C_{p}$, λ, and *h* were 6.05%, 13.86%, and 14.94%, respectively. Compared to sensor config:2, the errors in the identified material data were approximately doubled for each parameter. This may be due to the fact that the accurate identification of thermal conductivity λ requires more spatial gradient information on the temperature, which was not significantly present in sensor config:1. Consequently, the error in the uncertain λ propagated to other estimated parameters, such as $C_{p}$ and *h*. It is worth noting that, in the optimization procedure, the thermal conductivity λ always converged first (see, e.g., Figure 5a and Figure 6a).

Furthermore, the maximum error between the gPINN model and the measured temperature at the sensor locations was ΔT=0.03 [K], which was lower than the error for the config:2 arrangement. This is again due to the fact that, in config:1 (see Figure 2), the temperature sensor data did not contain a sufficient spatial gradient. As a result, the gPINN converged to a more uniform temperature distribution.

### 6.3. Effect of Noise

Measurement data always contains a certain level of noise, and introducing noise to the features or targets in both the forward and inverse problems can impact the final results. Therefore, it is important to perform an uncertainty quantification of the models to gain insights into the range of their applicability. To assess the robustness of the applied inverse gPINN for parameter identification, noise was introduced into the virtual temperature sensor data. The commonly used NiCr/Ni thermocouples have an application range between 0 ${}^{\circ }$C and 1100 ${}^{\circ }$C, with an accuracy of ±1.5 ${}^{\circ }$C. To evaluate the robustness of the inverse problem solution against noise, three different noise levels were considered.

The data was corrupted by adding Gaussian noise with a mean of zero and standard deviations of σ=0.1
${}^{\circ }$C, 0.5 ${}^{\circ }$C, and 1.5 ${}^{\circ }$C. The effect of the noise on the sensor data of config:2 was studied, as the parameters were estimated more accurately in this configuration compared to config:1, as shown earlier. Figure 7 shows the temperature sensor data (on the surface of the cylinder) with different levels of added noise. It is worth noting that the noise levels were much higher than the measurement error of the thermocouples (±1.5 ${}^{\circ }$C) considered, as a Gaussian noise with a standard deviation of σ can introduce errors up to 4σ to the data.

In order to identify the material data and assess their uncertainty, the gPINN optimization process was performed multiple times. Specifically, the optimization was restarted five times with different initial values for the model parameters, including the network, material, and cooling parameters. A total of 5,000,000 iterations were considered, and the optimization was stopped, even if the parameters had not fully converged. This approach took into account not only the accuracy of the estimated parameters, but also the efficiency of the algorithm in solving the inverse problem. To perform basic statistical analysis, the values of the estimated material parameters at the final iteration step were extracted for each restart. The mean and standard deviation of the estimated parameters were then calculated based on these values.

The results of the uncertainty quantification study are presented in Table 2. It is observed that, as the noise level increased, the uncertainty of the estimated parameters also increased. However, it should be noted that, due to fact that the optimization problem did not fully converge in some restarts (as a result of the fixed number of iterations set at 5,000,000), the accuracy of the estimated data was still acceptable, with the true value mostly falling within the error bars. It is found that, in principle, the gPINN has the capability to solve the inverse problem for any noise level and restart given more time (with additional iterations up to 20,000,000).

The convergence history of the five restarts for σ=0.0 and σ=1.5 is shown in Figure 8. It can be observed that, for the case without noise, each restart converged faster compared to the case with σ=1.5. It is important to note that, for the extreme case with σ=1.5, the maximum introduced error was around 6 ${}^{\circ }$C, which is nearly twice the maximum temperature difference between the sensors on the surface and center of the cylinder (see Figure 2d). Therefore, with such high uncertainty in the input measurement data, the optimization algorithm will take longer to find reasonable material parameters that accurately describe the temperature distribution in the sample. However, it can be concluded that the applied algorithm was robust against extreme noise levels, although it had lower efficiency in terms of the convergence speed (i.e., it required more GPU time to converge). To improve the convergence speed of the algorithm, it may be beneficial to start with reasonable initial parameter values instead of purely random initialization, as attempted in this work. Overall, physics-based machine learning approaches are generally more robust to noise compared to purely data-driven approaches. This is because physics-based models incorporate regularization constraints based on the underlying physics of the process. There are also other studies in the literature that demonstrate the noise robustness of PINNs, as mentioned in references [43,44,45].

## 7. Summary and Discussion

In this study, a gPINN model was utilized to solve and identify an air-cooling problem that is relevant to induction hardening processes. The effectiveness of the gPINN model was demonstrated by comparing its solution to the Finite Element Method (FEM) results in the context of the free air cooling of a cylindrical sample. The gPINN model showed a small residual error, thus indicating its successful performance in solving the forward problem.

Subsequently, the focus shifted to the inverse problem of the material parameter identification. FEM benchmark data were generated with known material data and boundary condition parameters, thereby serving as the input for the inverse problem. Two different configurations of virtual temperature sensors were considered for parameter identification using the gPINN model. One configuration involved varying axial and radial positions (config:2), while the other configuration focused on changing only the radial positions (config:1). The results demonstrated that both sensor configurations led to the accurate identification of the material parameters. However, the accuracy was higher when using sensor data from config:2, which suggests the importance of having a sufficient spatial resolution across the sample in the temperature measurements. Based on these findings, it is recommended to prioritize temperature sensor measurements that provide an adequate spatial resolution across the sample when conducting material parameter identification using the gPINN model.

The impact of noise on the accuracy of the estimated parameters was also examined in this study. Gaussian noise at three different levels was added to the virtual temperature sensor data. The results indicate that the presence of measurement noise did not have a significant effect on the accuracy of the estimated parameters. However, it increased the convergence time of the optimization process. It is worth noting that, in practical applications, the effect of white noise can be mitigated by implementing an appropriate low-pass filter or by initializing the optimization with reasonable initial values for the unknown parameters.

In this study, a short time temperature history equivalent to a temperature drop of 25 [K] was considered for estimating the material data. A cooling to room temperature would take approximately 500 [s] for the considered material data of this study. However, the entire cooling time was not included in the inverse model for a couple of reasons. Firstly, the entire cooling time would significantly increase the training time due to the requirement of a large number of collocation points in the physical domain for continuous time models [14]. Additionally, GPU memory limitations (15 GB available on the machine) make it challenging to perform such optimizations with a high number of collocation points in both space (2D) and time.

Another important consideration is the balance between the number of temporal and spatial samples. If the spatial resolution of the temperature is not significantly large, using long-time temperature simulations in the PINN model may lead to convergence towards an effective lumped mass model, thereby resulting in inaccurate estimates of the material data. For long-time simulations, it is advisable to gradually proceed in time and propagate the solution or to utilize PINNs with discrete time models [46].

Due to the nature of the governing equations, it is not possible to have three unknowns in the equations governing the boundary conditions. Only two ratios of the unknown parameters coupled to the boundary condition equations can be identified. Furthermore, it was found that a 10 [s] cooling history was sufficient for identifying the thermo-physical material data when the emissivity coefficient was already known.

For practical applications, to apply the approach to an induction heating test rig, one has to first heat the sample up to achieve a uniform or close-to-uniform temperature distribution across the sample. This can be achieved by applying a low-frequency power source that thoroughly heats the sample. In scenarios where thorough hardening is not possible, such as with very large samples, the proper selection of the spatial positions for the temperature sensors becomes crucial in order to accurately capture the temperature distribution in the sample and, importantly, to estimate the material data accurately. Therefore, employing the applied approach for material data identification required temperature acquisition in different locations that correspond to either the config:1 or config:2 sensor arrangements. Based on this approach, it was necessary to drill the sample and record the temperature at the desired locations. We showed that, with five sensors, it was possible to reconstruct the temperature field in the sample. However, further study and exploration using PINNs is necessary to assess the feasibility of the material data identification when using a reduced number of sensors (e.g., two or only one sensor) or when solely relying on surface temperature information without the need for drilling and interfering with the sample.

In this work, the assumption was made that the material properties were constant. However, in reality, material properties are often temperature-dependent and can vary over a wide range, especially in processes such as induction hardening. Therefore, to apply the approach in practical applications, an upgrade is needed to estimate temperature-dependent material properties. Two directions are suggested for further investigation in this regard. The first direction involves applying the current approach to different windows of the cooling history, such as temperature drops of 5 to 10 °C. By applying the PINN model to each temperature window, approximate material data can be obtained by assuming them to be constant within that specific temperature range. The second strategy involves designing a new PINN architecture that can handle temperature-dependent material properties. Instead of assuming the material properties as constant parameters in the PINN model, they can be treated as functions of the temperature that are represented by neural networks. For example, such an architecture can account for $C_{p}=NN_{1}\left(T\right)$, $\lambda =NN_{2}\left(T\right)$, and $h=NN_{3}\left(T\right)$, where $NN_{i}$ represents the approximating functions represented by neural networks. These networks can either share the same weights and biases or have different architectures. Further study and investigation are required to evaluate the performance of such a PINN model. Future research can be conducted to explore these possibilities.

## Figures and Tables

**Figure 1 materials-16-05013-f001:**
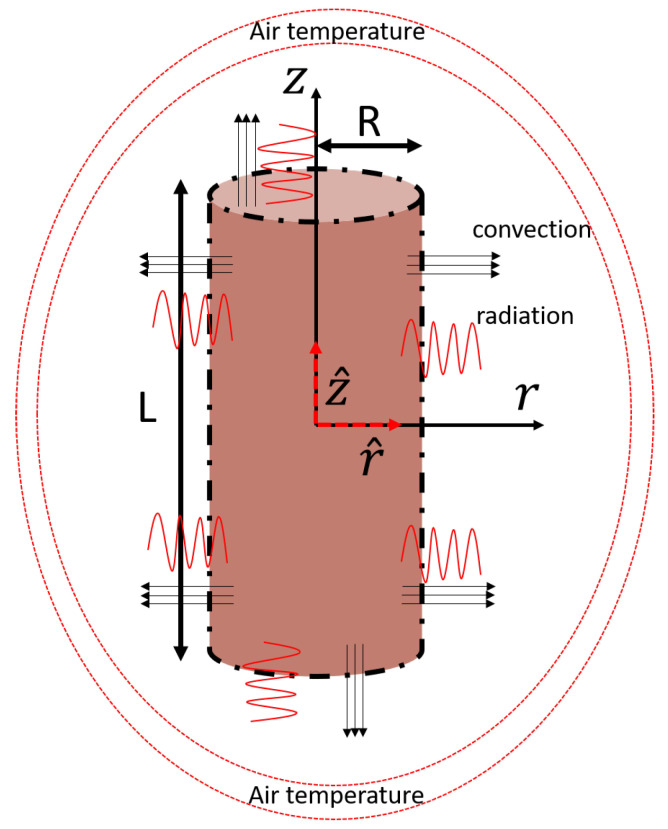
Schematic of a cylindrical sample undergoing free air cooling, with illustrating heat load directions on the boundaries. The length of the cylinder is denoted by L, and the radius is denoted by R. The sample has an initial uniform temperature of $T_{init}$. Cooling via radiation and free convection takes place along the directions of the surface normals, which are indicated by vectors $\hat{r}$, $\hat{z}$, and $-\hat{z}$.

**Figure 2 materials-16-05013-f002:**
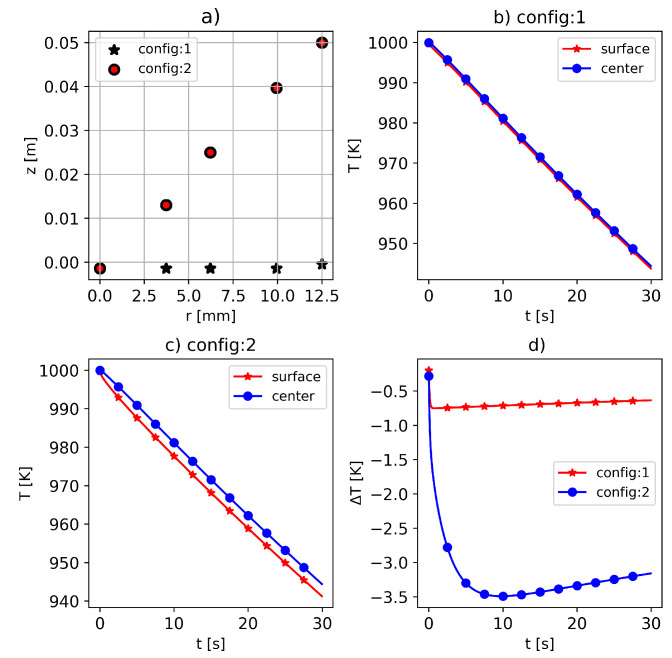
(**a**) Position of virtual sensors. In sensor configuration one (config:1), the sensors are distributed radially, while in config:2, the axial position of sensors is varied. Temperature evolution for sensors located on the surface and center of the sample in (**b**) config:1 and (**c**) config:2. (**d**) Temperature difference between the sensors at the surface and center of the cylindrical sample for both sensor configurations.

**Figure 3 materials-16-05013-f003:**
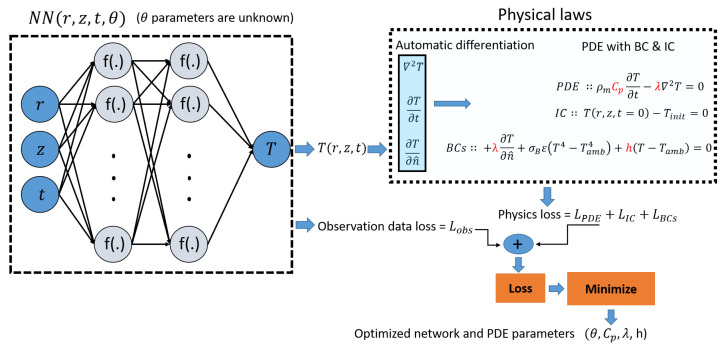
Schematics of the applied PINN model structure.

**Figure 4 materials-16-05013-f004:**
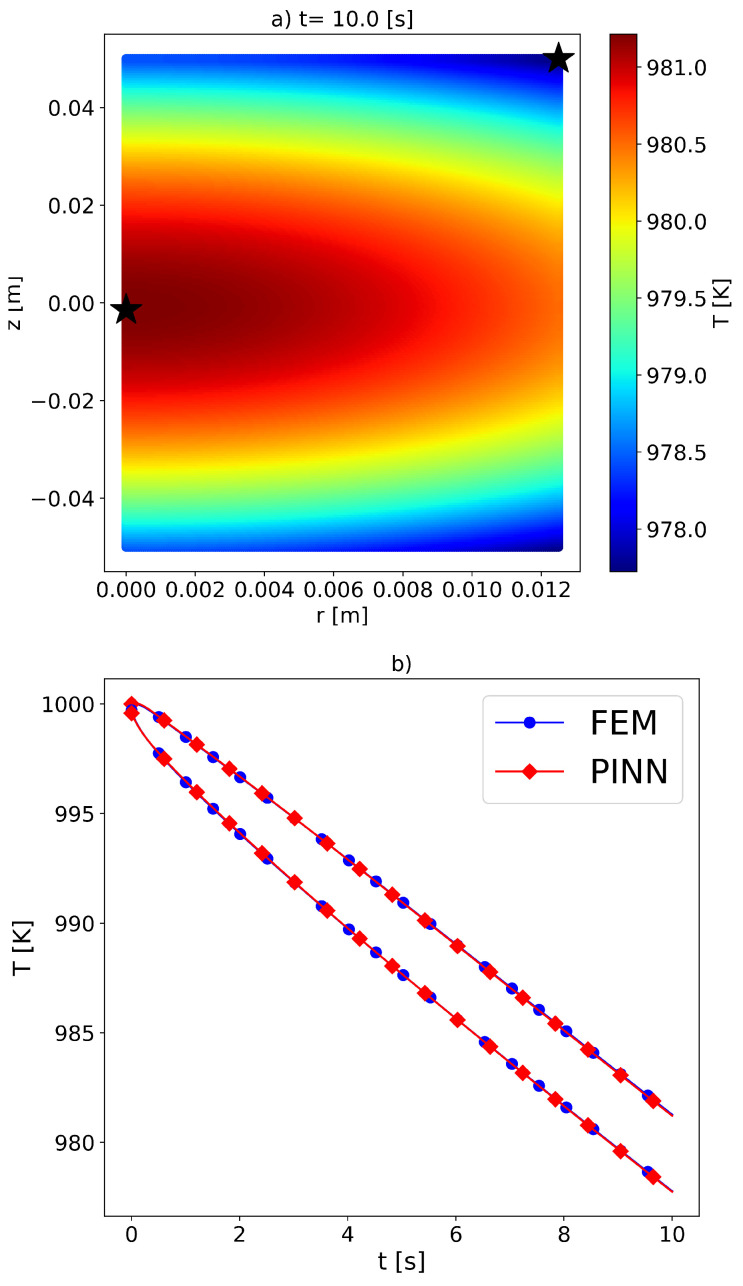
Forward solution of the PDE equation using PINNs. (**a**) Temperature distribution in the r–z plane at the time frame t = 10 [s]. (**b**) Comparison of temperature evolution between FEM and PINN solutions for two sensor locations (marked with star symbols in panel (**a**)).

**Figure 5 materials-16-05013-f005:**
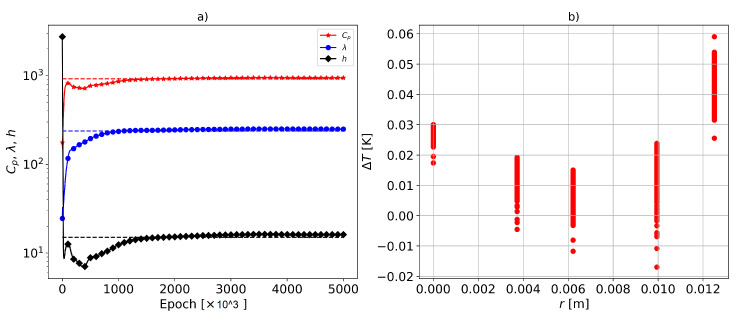
(**a**) Identified parameters compared to the true values shown by horizontal dashed lines. The relative errors of the identified parameters $C_{p}$, λ, and *h* were 2.6%, 5.27%, and 7.35%, respectively. (**b**) The temperature difference between the predicted and true values generated using FEM at the sensor locations. The vertical variation of the error at a given sensor location corresponds to different time steps. The sensor configuration is config:2.

**Figure 6 materials-16-05013-f006:**
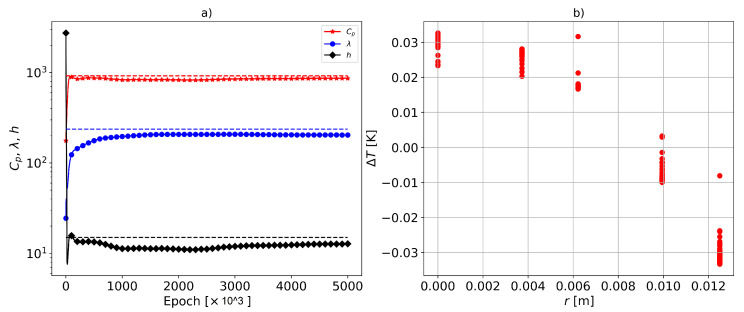
(**a**) Identified parameters compared to the true values shown by horizontal dashed lines. The relative errors of the identified parameters $C_{p}$, λ and *h* were 6.05%, 13.86%, and 14.94% respectively. (**b**) The temperature difference between the predicted and true one generated with FEM at the sensor locations. The vertical variation of the error at a given sensor location corresponds to different time steps. The sensor configuration is config:1.

**Figure 7 materials-16-05013-f007:**
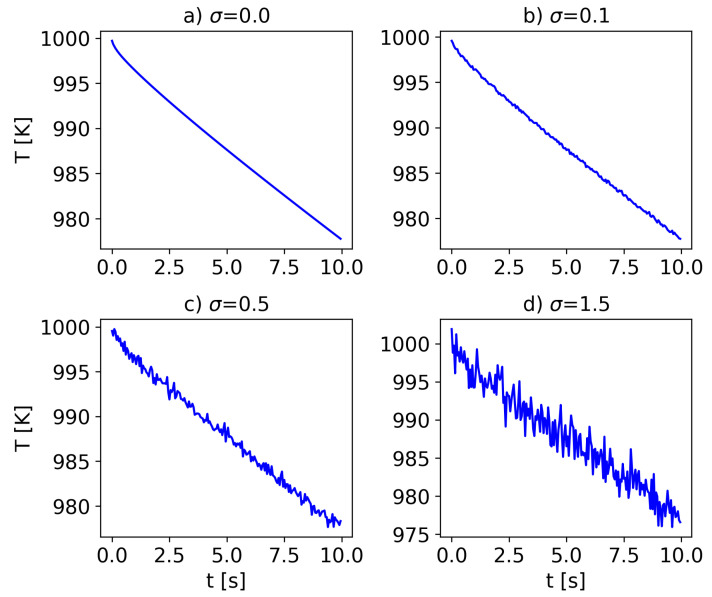
Temperature data with different levels of added Gaussian noise. The noise has a mean of zero and standard deviation of σ, where σ values were the following: (**a**) 0.0, (**b**) 0.1, (**c**) 0.5, and (**d**) 1.5.

**Figure 8 materials-16-05013-f008:**
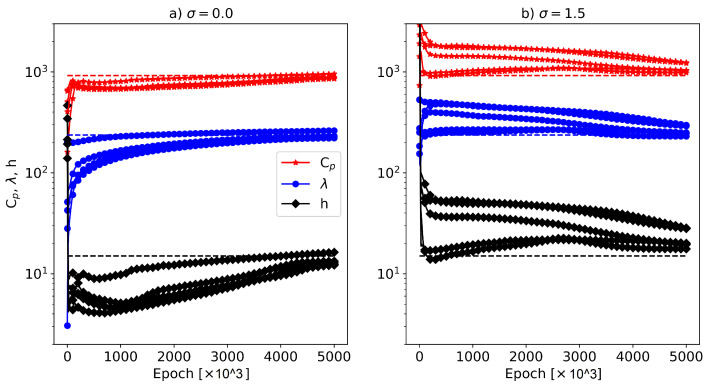
Convergence history of the parameters for each optimization restart for two cases: (**a**) without added Gaussian noise (σ=0.0) and (**b**) with added Gaussian noise (σ=1.5).

**Table 1 materials-16-05013-t001:** Material and process data for the Finite Element Method (FEM) solution of the cooling process.

$\rho_{m}$ [kg/m${}^{3}$]	$C_{p}$ [J/kg·K]	λ [W/m·K]	*h* [W/m${}^{2}$·K]	ϵ	$T_{\mathrm{init}}$ [K]	$T_{\mathrm{amb}}$ [K]
2700	921	237	15	0.32	1000	293.15

**Table 2 materials-16-05013-t002:** Accuracy of the identified material data in the presence of added Gaussian noise with a mean of zero and different levels of standard deviation (σ). The effect of noise was studied for sensor data of config:2. The true values of the material data are reported in Table 1.

σ [${}^{\circ }$C]	$C_{p}$ [J/kg·K]	λ [W/m·K]	*h* [W/m${}^{2}$·K]
0.0	886.25 ±36.61	233.56 ±14.60	13.45 ±1.52
0.1	940.02 ±57.30	249.77 ±21.35	15.87 ±2.42
0.5	1259.03 ±90.88	335.96 ±24.43	30.02 ±3.98
1.5	1102.99 ±106.21	263.38 ±26.53	22.76 ±4.57

## Data Availability

Data are available from the corresponding author upon a reasonable request.

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
