# Peer review of "Material Data Identification in an Induction Hardening Test Rig with Physics-Informed Neural Networks"

_materials, 2023, doi:10.3390/ma16145013_

Round 1
Reviewer 1 Report
Dear Editor: For the study to be accepted, the author (s) must respond to the following comments point by point to reach the level of a high-quality journal.
1. The followings are my comments regarding this manuscript. The authors should briefly discuss their innovation in the abstract, which has to be improved with more information.
2. Missing the test procedure, explaining the calculations, and is disorganized. The abstract needs to be improved by highlighting this paper's key findings.The entire abstract section must be revised to briefly explain this research study's importance, investigations, and outcomes with advantages/significance.
Abstract :
3. The entire abstract section must be revised to briefly explain this research study's importance, investigations, and outcomes with advantages/significance.
Introduction:
4. The introduction section is not up to the mark. In the introduction section, you only need to connect state-of-the-art to your paper goals. Hence modify the entire section accordingly and present the specific goals/research objectives in the last part of the introduction section.
5. Why is this study important? Why is modeling important for material behavior? What scientific problems are the authors going to address?
6. MAJOR comment: The authors requested and must add more information and supported studies to the introduction since the introduction is poor and needs to be strengthened. The authors did not use the last decade's papers on the effect of using neural networks on solving the problem in different research areas with multiple functional parameters!! The following papers published in reputed journals must be used in the introduction and the results in these papers have to be compared with the outcomes of this study to enhance the quality of the manuscript:
· The effectiveness of ensemble-neural network techniques to predict peak uplift resistance of buried pipes in reinforced sand
· Artificial neural networks (ANN), MARS, and adaptive network-based fuzzy inference system (ANFIS) to predict the stress at the failure of concrete with waste steel slag coarse …
· A new auto‑tuning model for predicting the rock fragmentation: a cat swarm optimization algorithm
· Metamodel techniques to estimate the compressive strength of UHPFRC using various mix proportions and a high range of curing temperatures
· Introducing stacking machine learning approaches for the prediction of rock deformation
· Stacking Ensemble Tree Models to Predict Energy Performance in Residential Buildings
· Early prediction of COVID-19 outcome using artificial intelligence techniques and only five laboratory indices
Methodology:
1. A flowchart should be provided for the work process. The flow chart of the study has to be described in the steps.
2. Many grammatical errors need to be corrected. Several grammar errors can be observed in the paper, which is negatively affected by the paper's quality. Using " we, our …….." not allowed in scientific article writing?
3. What are the limitations of this work? Please mention them in the discussion and conclusion sections.
Dear Editor: For the study to be accepted, the author (s) must respond to the following comments point by point to reach the level of a high-quality journal.
1. The followings are my comments regarding this manuscript. The authors should briefly discuss their innovation in the abstract, which has to be improved with more information.
2. Missing the test procedure, explaining the calculations, and is disorganized. The abstract needs to be improved by highlighting this paper's key findings.The entire abstract section must be revised to briefly explain this research study's importance, investigations, and outcomes with advantages/significance.
Abstract :
3. The entire abstract section must be revised to briefly explain this research study's importance, investigations, and outcomes with advantages/significance.
Introduction:
4. The introduction section is not up to the mark. In the introduction section, you only need to connect state-of-the-art to your paper goals. Hence modify the entire section accordingly and present the specific goals/research objectives in the last part of the introduction section.
5. Why is this study important? Why is modeling important for material behavior? What scientific problems are the authors going to address?
6. MAJOR comment: The authors requested and must add more information and supported studies to the introduction since the introduction is poor and needs to be strengthened. The authors did not use the last decade's papers on the effect of using neural networks on solving the problem in different research areas with multiple functional parameters!! The following papers published in reputed journals must be used in the introduction and the results in these papers have to be compared with the outcomes of this study to enhance the quality of the manuscript:
· The effectiveness of ensemble-neural network techniques to predict peak uplift resistance of buried pipes in reinforced sand
· Artificial neural networks (ANN), MARS, and adaptive network-based fuzzy inference system (ANFIS) to predict the stress at the failure of concrete with waste steel slag coarse …
· A new auto‑tuning model for predicting the rock fragmentation: a cat swarm optimization algorithm
· Metamodel techniques to estimate the compressive strength of UHPFRC using various mix proportions and a high range of curing temperatures
· Introducing stacking machine learning approaches for the prediction of rock deformation
· Stacking Ensemble Tree Models to Predict Energy Performance in Residential Buildings
· Early prediction of COVID-19 outcome using artificial intelligence techniques and only five laboratory indices
Methodology:
1. A flowchart should be provided for the work process. The flow chart of the study has to be described in the steps.
2. Many grammatical errors need to be corrected. Several grammar errors can be observed in the paper, which is negatively affected by the paper's quality. Using " we, our …….." not allowed in scientific article writing?
3. What are the limitations of this work? Please mention them in the discussion and conclusion sections.
Author Response
We sincerely appreciate the time and effort the reviewers have put into carefully reading our manuscript. Their insightful comments and suggestions have significantly improved the quality of the manuscript. We have incorporated all the recommended changes and additions, which are now mostly highlighted in blue font throughout the revised version of the manuscript. In the following sections, we address each of the reviewers' comments and suggestions in detail.
Reviewer 1:
- The followings are my comments regarding this manuscript. The authors should briefly discuss their innovation in the abstract, which has to be improved with more information.
- Missing the test procedure, explaining the calculations, and is disorganized. The abstract needs to be improved by highlighting this paper's key findings. The entire abstract section must be revised to briefly explain this research study's importance, investigations, and outcomes with advantages/significance.
Abstract :
- The entire abstract section must be revised to briefly explain this research study's importance, investigations, and outcomes with advantages/significance.
Thank you for the suggestion. We have revised and rephrased the abstract to improve its clarity.
Introduction:
- The introduction section is not up to the mark. In the introduction section, you only need to connect state-of-the-art to your paper goals. Hence modify the entire section accordingly and present the specific goals/research objectives in the last part of the introduction section.
Thanks for the suggestion. The introduction has been revised and new relevant literature is added.
- Why is this study important? Why is modeling important for material behavior? What scientific problems are the authors going to address?
The study investigates the feasibility of identifying material data through Physics-informed Neural Networks (PINNs) using a limited number of temperature sensor data points in a 2D transient heat transfer problem. Specifically, the focus is on a cooling process relevant to induction hardening. Accurate identification of material data, including the specific heat, thermal conductivity, and heat convection coefficient, is of paramount importance in understanding and optimizing heat transfer processes.
By utilizing benchmark data, the study demonstrates the ability to infer material data accurately, even in the presence of measurement noise. This capability opens up opportunities for practical applications in various industries, including materials engineering, manufacturing, and heat treatment processes. Accurate material data identification enables engineers and researchers to make informed decisions, optimize designs, and improve the efficiency and performance of heat transfer systems.
Furthermore, the study contributes to the advancement of the field by demonstrating the effectiveness of PINNs in material data identification. PINNs offer a promising approach to solving complex inverse problems in various scientific and engineering domains. The ability to accurately estimate material properties using limited sensor data showcases the potential of PINNs as a valuable tool in data-driven engineering applications.
We incorporated part of this reply into the introduction.
- MAJOR comment: The authors requested and must add more information and supported studies to the introduction since the introduction is poor and needs to be strengthened. The authors did not use the last decade's papers on the effect of using neural networks on solving the problem in different research areas with multiple functional parameters!! The following papers published in reputed journals must be used in the introduction and the results in these papers have to be compared with the outcomes of this study to enhance the quality of the manuscript:
- The effectiveness of ensemble-neural network techniques to predict peak uplift resistance of buried pipes in reinforced sand
- Artificial neural networks (ANN), MARS, and adaptive network-based fuzzy inference system (ANFIS) to predict the stress at the failure of concrete with waste steel slag coarse …
- A new auto‑tuning model for predicting the rock fragmentation: a cat swarm optimization algorithm
- Metamodel techniques to estimate the compressive strength of UHPFRC using various mix proportions and a high range of curing temperatures
- Introducing stacking machine learning approaches for the prediction of rock deformation
- Stacking Ensemble Tree Models to Predict Energy Performance in Residential Buildings
- Early prediction of COVID-19 outcome using artificial intelligence techniques and only five laboratory indices
Thank you for the feedback. We appreciate your suggestion, but after careful consideration, we believe that the papers you mentioned may not be directly relevant to the specific topic addressed in our work. We have made sure to include citations to the most relevant and recent works on PINN models throughout the manuscript. However, we revisited our references to ensure that we have not overlooked any important papers.
Methodology:
- A flowchart should be provided for the work process. The flow chart of the study has to be described in the steps.
Thank you for your feedback. We appreciate your suggestion, but we do not find it relevant to the topic of our work.
- Many grammatical errors need to be corrected. Several grammar errors can be observed in the paper, which is negatively affected by the paper's quality. Using " we, our …….." not allowed in scientific article writing?
Thank you for bringing that to our attention. We apologize for any language issues in the paper. We have made significant improvements to the English language throughout the manuscript to ensure clarity and readability.
- What are the limitations of this work? Please mention them in the discussion and conclusion sections.
We have added a section in the Discussion to address the limitations of the proposed approach. We acknowledge that every method has its limitations, and it is important to discuss them to provide a comprehensive understanding of the study. The limitations highlight the areas where the proposed approach may not be suitable or may require further development. We believe that by addressing these limitations, future research can focus on overcoming them and improving the applicability of the approach. Thank you for suggesting this addition.
Reviewer 2 Report
The article develops a physics informed neural network (PINN) algorithm for predicting material properties and heat transfer variables from sample temperatures. Overall, the text is interesting and the article contains all the necessary elements. Certain improvements are necessary in the analysis of results and discussion, and the authors should refer to the comments below to improve those sections:
1. Line 72: "They Stefan problem..."
2. Line 80: I think that "the 3D (two spatial and one temporal dimension)" should be called transient two-dimensional problem or time-dependent two-dimensional problem.
3. Figure 2-b) and c): the center and surface temperatures are hardly distinguishable from each other. I suggest using different line colors instead of point markers. Also, figure 2-c) shows that the center of the cylinder sample (red star marker) is cooling faster than its surface (red rounded marker), whereas both center and surface starts cooling at the same temperature (Tinit = 1000 K for t = 0 seconds). This is correct?
4. Figure 2-b) (configuration 1) shows no difference between center and surface points which is expected since aluminum is a material with high thermal conductivity (λ = 237 W/mK). However, carbon steel (λ = 45 W/mK) or stainless steel (λ = 15 W/mK) have lower thermal conductivities and could produce some temperature difference between sample center and surface. The temperature distribution across the sample depends on the material type (cp, λ, ρ) and heat transfer variables (h, ϵ): how do they influence the PINN prediction force?
5. By selecting different sample geometries, or different sample materials and heat transfer variables the NN properties (layer depth and width, loss weights, Adam optimizer value) should be chosen differently or not? How flexible is the PINN approach?
6. You generate temperature data using FEM analysis for the PINN model. In practice, how should this work? A material engineer working on metal hardening could estimate the material properties only by measuring surface temperatures or should also drill test samples to measure the center temperatures?
7. Generally, material properties are temperature-dependent, especially noticeable across large temperature differences such as in hardening process. In the present study you select constant values for material properties. How could the PINN model be upgraded to account for temperature-dependent material properties?
8. Figures 5-a) and 6-a) report the comparison between predicted and true cp, λ and h on a logarithmic scale, which is hard to quantify visually. How much is the relative difference (in %) or the uncertainty (like in table 2) between the predicted and true values? The labels for figures 5 and 6 should state the sensor configuration.
9. Table 2 shows that the relative uncertainty is the highest for an added noise level of σ = 0.5 °C (around 30%)? What happens with the PINN at σ = 0.5 °C, why is the prediction error higher than than obtained with a larger temperature noise of σ = 1.5 °C? Line 317: the PINN algorithm can be considered robust also for the noise level of σ = 0.5 °C? Line 338: in my opinion the noise level of σ = 0.5 °C significantly affects the predicted parameters. Please comment in the text this issue.
10. Figure 8: add the legend for the different color markers.
11. Did you compare your results with similar studies from the literature, especially in that concerning the PINN prediction capability at different noise levels?
English language and writing style are fine, minor text editing required and a check for typos.
Author Response
We sincerely appreciate the time and effort the reviewers have put into carefully reading our manuscript. Their insightful comments and suggestions have significantly improved the quality of the manuscript. We have incorporated all the recommended changes and additions, which are now mostly highlighted in blue font throughout the revised version of the manuscript. In the following sections, we address each of the reviewers' comments and suggestions in detail.
Comments and Suggestions for Authors
The article develops a physics informed neural network (PINN) algorithm for predicting material properties and heat transfer variables from sample temperatures. Overall, the text is interesting and the article contains all the necessary elements. Certain improvements are necessary in the analysis of results and discussion, and the authors should refer to the comments below to improve those sections:
- Line 72: "They Stefan problem..."
Thanks for the notice. It is corrected.
- Line 80: I think that "the 3D (two spatial and one temporal dimension)" should be called transient two-dimensional problem or time-dependent two-dimensional problem.
Thank you for the suggestion. We have made the necessary corrections, and the manuscript now consistently addresses the 2D transient heat transfer problem throughout.
- Figure 2-b) and c): the center and surface temperatures are hardly distinguishable from each other. I suggest using different line colors instead of point markers. Also, figure 2-c) shows that the center of the cylinder sample (red star marker) is cooling faster than its surface (red rounded marker), whereas both center and surface starts cooling at the same temperature (Tinit = 1000 K for t = 0 seconds). This is correct?
Thank you for the suggestion. We have updated the figure and added a new panel to show the temperature difference between the surface and the center for both sensor configurations. We acknowledge the observation that the surface cools down faster than the center. Furthermore, we apologize for the mistake in the legend, which has now been corrected.
- Figure 2-b) (configuration 1) shows no difference between center and surface points which is expected since aluminum is a material with high thermal conductivity (λ = 237 W/mK). However, carbon steel (λ = 45 W/mK) or stainless steel (λ = 15 W/mK) have lower thermal conductivities and could produce some temperature difference between sample center and surface. The temperature distribution across the sample depends on the material type (cp, λ, ρ) and heat transfer variables (h, ϵ): how do they influence the PINN prediction force?
Thank you for your input. The resolution of temperature measurements does play a role in the accuracy of material data identification using the PINN model. In this study, we carefully selected combinations of material data to include both low and moderate temperature resolutions based on the sensor arrangements. However, it is important to note that the proposed method is applicable to any material and cooling conditions. We have addressed your question by integrating it into the text at line 188.
- By selecting different sample geometries, or different sample materials and heat transfer variables the NN properties (layer depth and width, loss weights, Adam optimizer value) should be chosen differently or not? How flexible is the PINN approach?
Thank you for your question. In principle, the PINN approach is flexible and can accommodate variations in material data or heat transfer variables. However, when applying the approach to different geometries, it is important to carefully consider the sampling of the solution domain governed by the partial differential equations (PDEs). The selection of collocation points, a crucial hyperparameter in PINN training, can be done using various sampling strategies. Depending on the specific problem, a suitable sampling strategy should be chosen.
Furthermore, for different geometries, other hyperparameters of PINNs such as layer depth, width, loss term weights, and optimizer settings should be optimized accordingly. These hyperparameters may need to be adjusted to achieve optimal performance and accurate material data identification for specific geometries.
We have incorporated some discussion regarding your question in the manuscript, at line 268, and line 302.
- You generate temperature data using FEM analysis for the PINN model. In practice, how should this work? A material engineer working on metal hardening could estimate the material properties only by measuring surface temperatures or should also drill test samples to measure the center temperatures?
Employing the applied approach for material data identification requires temperature acquisition in different locations corresponding to one of the config:1 and config:2 sensor arrangements. Therefore, it is necessary to drill the sample and record the temperature. However, further study and exploration using PINNs is necessary to check the feasibility of material data identification using few sensor data (for example two or one sensor) or just using surface temperature information without interfering with the sample by drilling.
We have this discussion to the conclusion section.
- Generally, material properties are temperature-dependent, especially noticeable across large temperature differences such as in hardening process. In the present study you select constant values for material properties. How could the PINN model be upgraded to account for temperature-dependent material properties?
The applied approach can be utilized to analyze different windows of cooling history, such as temperature drops ranging from 5 to 10 °C. By applying the approach within these specific temperature ranges, it is possible to approximate the material data corresponding to those conditions.
Furthermore, the flexibility of the PINN framework allows for the incorporation of temperature-dependent material data identification through the selection of appropriate PINN architectures. Instead of considering the material data as constant parameters in the PINN model, they can be represented as functions of temperature using neural networks. However, further investigation and study are needed to evaluate the performance of such a PINN model in capturing temperature-dependent material behavior. The authors express their interest in conducting such studies in future research endeavors.
We have added this discussion to the conclusion and discussion section.
- Figures 5-a) and 6-a) report the comparison between predicted and true cp, λ and h on a logarithmic scale, which is hard to quantify visually. How much is the relative difference (in %) or the uncertainty (like in table 2) between the predicted and true values? The labels for figures 5 and 6 should state the sensor configuration.
Thanks for the suggestion. We added the relative error to the description of figures and in the caption. Furthermore, the relevant configurations is added to the captions of the figures.
- Table 2 shows that the relative uncertainty is the highest for an added noise level of σ = 0.5 °C (around 30%)? What happens with the PINN at σ = 0.5 °C, why is the prediction error higher than that obtained with a larger temperature noise of σ = 1.5 °C? Line 317: the PINN algorithm can be considered robust also for the noise level of σ = 0.5 °C? Line 338: in my opinion the noise level of σ = 0.5 °C significantly affects the predicted parameters. Please comment in the text this issue.
We utilized random initialization for the starting parameters, including both PDE parameters and neural network (NN) parameters. Our investigation reveals that the noise primarily impacts the convergence time of the optimization process. Specifically, for the case where \sigma=0.5, we observed that the optimization was interrupted and incomplete during the final iteration, as depicted in Figure [1]. Additionally, for the same \sigma=0.5 case, the combination of initial parameter values was not ideal, resulting in a longer convergence time to reach the optimum solution. This observation was not initially apparent since post-processing, involving averaging and uncertainty calculation, was automatically performed by a Python script without visualizing the optimization history.
We have now updated Table 2 to reflect the changes. The updated table demonstrates an increase in prediction uncertainty as the level of noise added to the data increases.
Figure 1 Optimization history for different noise levels
In principle, conducting a thorough uncertainty quantification would involve extending the optimization process to a higher number of iterations, typically in the range of 10-20 million, and repeating the restart procedure multiple times, preferably 10 or more. However, in this work, we have chosen to avoid such an extensive optimization process due to its low computational efficiency. Instead, we focused on starting from reasonable initial parameter values, which has proven to significantly reduce computational time while still yielding meaningful results.
We added a sentence regarding this comment in line 432.
- Figure 8: add the legend for the different color markers.
Thanks for the suggestion. The legends are added.
- Did you compare your results with similar studies from the literature, especially in that concerning the PINN prediction capability at different noise levels?
Physics based machine learning is more robust to noise compared to pure data driven approaches. This is due to the incorporated constraints originating from the correct physics of the process. There is some literature which also show and report noise robust estimates from PINN for example in [45-47]
We added a sentence regarding this comment and cited the relevant papers on this topic at line 437.
Reviewer 3 Report
The paper is interesting for the readers and can contribute in solving ill-posed problems in the scope of mathematical physics. I have some questions and recommendations to the authors.
1) Justify, please, the choice of the weights in Formula (4).
2) Schematics of the applied PINN model shown in Fig.3 is not clear enough.
3) The comma in Line 253 is redundant, in my opinion.
4) I think that the dynamic random sampling strategy is not the most effective one when solving inverse problems for partial differential equations or systems of parabolic end elliptic type. There are exist the so called Carleman conditions ensuring the uniqueness of the solutions to such problems.
5) Results shown in Fig. 5b need to be described more accurately. I also recommend the authors to change the capture of this Fig: " here are the virtual sensor data ...".
6) Fig. 7 should be renamed. The quality of this Figure is also not very good.
English Language is well.
Author Response
We sincerely appreciate the time and effort the reviewers have put into carefully reading our manuscript. Their insightful comments and suggestions have significantly improved the quality of the manuscript. We have incorporated all the recommended changes and additions, which are now mostly highlighted in blue font throughout the revised version of the manuscript. In the following sections, we address each of the reviewers' comments and suggestions in detail.
Comments and Suggestions for Authors
The paper is interesting for the readers and can contribute in solving ill-posed problems in the scope of mathematical physics. I have some questions and recommendations to the authors.
1) Justify, please, the choice of the weights in Formula (4).
Imbalanced gradients in multi-objective optimization are a well-known problem, particularly for PINN models. These imbalanced gradients can arise from differences in the units of measurement for the loss terms or variations in the number of samples used to calculate them. As a result, during optimization, the terms with the highest magnitude receive more attention. Several approaches have been proposed to address this issue.
In our work, we have utilized a version of PINNs, which incorporates the gradients of the PDEs into the loss function. This modification significantly enhances performance and convergence compared to standard PINNs without adaptive weights.
Moreover, in the inverse transient heat problem considered in this study, there are relatively few observation sensor data points available (5*200=1000) compared to the 16500 collocation points used to sample the physical domain. To balance the magnitude of the loss terms, we have assigned a larger weight (10000) to the loss on the observation data. While this weight may not be optimized specifically for this problem, it has yielded satisfactory results in our study, with the estimated parameters showing good agreement with the true values.
We have included a revised discussion of this choice, beginning at line 247 of the manuscript.
2) Schematics of the applied PINN model shown in Fig.3 is not clear enough.
Thank you for the suggestion. We have made edits to the schematics in order to improve clarity and readability.
3) The comma in Line 253 is redundant, in my opinion.
Thanks for the notice, it is corrected now.
4) I think that the dynamic random sampling strategy is not the most effective one when solving inverse problems for partial differential equations or systems of parabolic end elliptic type. There are exist the so called Carleman conditions ensuring the uniqueness of the solutions to such problems.
We acknowledge that there are various types of sampling strategies that can be employed when solving PDEs using PINNs. However, the effectiveness and accuracy of each sampling approach may vary depending on the specific problem. In our feasibility study, we found that the chosen random sampling approach yielded satisfactory results, as the identified parameters were comparable to the true values.
It is important to note that in our study, we performed random sampling after 5000 iteration steps. With this strategy, if a sufficient number of samples are used, it is possible to explore a significant portion of the physical domain.
We have made the necessary adjustments to the sentence in line 302 to avoid any potential misinformation.
5) Results shown in Fig. 5b need to be described more accurately. I also recommend the authors to change the capture of this Fig: " here are the virtual sensor data ...".
Thank you for the suggestion. We have revised the figure to provide a clearer explanation. Additionally, we have made modifications to the caption to enhance its clarity.
6) Fig. 7 should be renamed. The quality of this Figure is also not very good.
Thanks for the suggestion. The caption has been revised and enhanced for better clarity.
Reviewer 4 Report
The study applies physics-informed neural networks (PINNs) to understand their capabilities of materials data identification when applied on an induction hardening test rig. Using heat equations and sensor data, the authors estimated the thermo-physical properties using PINNs and FEM. Their model can identify the material data using a few sensor data. The study is further checked for its applicability to the real measurement data of an air-cooled cylindrical sample. However, after carefully reviewing the paper, I have found some significant concerns that should be addressed.
- The plots in Figure 2 show the temperature data of sensors only until ~900(K). This indicates that the authors selected mostly linear portion of the temperature. It will be interesting to see the model's performance based on the data until room temperature. Authors should provide proper justification why only the linear portion of the temperature is selected for training the PINNs.
- Hyperparameter tuning is critical in identifying an optimal neural network architecture. Authors should elaborate on this and show why particular learning rates, depths, and widths of NN are selected. The authors should show some results that indicate the selected parameters yield the optimal results with no overfitting observed. The results should be shown on test data which is never used for training the model.
- The literature survey on PINNs seems incomplete. The statement in line 77-79, "This previous studies..." is very general. Authors statement of PINNs for 3-D heat transfer is not convincing. Transient axis-symmetric heat transfer problems may be appropriate.
- It is recommended to include an illustration showing the various stages of the induction hardening process should be provided. All the boundary conditions and heat-loading directions should be shown.
- The statement on line 58, "Artificial neural networks....invention in 1958..." is misleading as Frank Rosenblatt used a single feed-forward neuron "perceptron". Also, citations should be provided for various machine-learning tasks.
Author Response
We sincerely appreciate the time and effort the reviewers have put into carefully reading our manuscript. Their insightful comments and suggestions have significantly improved the quality of the manuscript. We have incorporated all the recommended changes and additions, which are now mostly highlighted in blue font throughout the revised version of the manuscript. In the following sections, we address each of the reviewers' comments and suggestions in detail.
Comments and Suggestions for Authors
The study applies physics-informed neural networks (PINNs) to understand their capabilities of materials data identification when applied on an induction hardening test rig. Using heat equations and sensor data, the authors estimated the thermo-physical properties using PINNs and FEM. Their model can identify the material data using a few sensor data. The study is further checked for its applicability to the real measurement data of an air-cooled cylindrical sample. However, after carefully reviewing the paper, I have found some significant concerns that should be addressed.
- The plots in Figure 2 show the temperature data of sensors only until ~900(K). This indicates that the authors selected mostly linear portion of the temperature. It will be interesting to see the model's performance based on the data until room temperature. Authors should provide proper justification why only the linear portion of the temperature is selected for training the PINNs.
Thank you for your comment. While there is indeed a subtle nonlinearity in the temperature data, it may not be visible to the naked eye. In our study, we considered a temperature history of 10 seconds, which corresponds to a temperature drop of 25 K. It would take around 500 seconds for the temperature to drop to room temperature. However, we did not include the entire cooling time in the inverse model for a couple of reasons.
Firstly, including the entire cooling time would significantly increase the training time, as continuous time models require a large number of collocation points in the physical domain [14]. Secondly, due to limitations in GPU memory (15 GB available on our machine), it is not easily feasible to perform such optimizations with a high number of collocation points in both spatial (2D) and temporal dimensions.
Another important consideration is the balance between the number of spatial and temporal samples. When the spatial resolution of temperature is not significantly large, using long time simulations in the PINN model may result in convergence to an effective lumped mass model, leading to incorrect estimation of material data. For long time simulations, it is advisable to gradually propagate the solution in time or use PINNs with discrete time models [48].
In conclusion, for the purpose of identifying the thermo-physical material data, we found that a 10-second cooling history is sufficient when the emissivity coefficient is already known. This is because the nonlinear heat flow term on the boundary is fixed. Due to the nature of the governing equations, it is not possible to have three unknowns in the equations governing the boundary conditions. Therefore, it is only feasible to identify two ratios of the unknown parameters coupled to the boundary condition equations.
We have discussed this point in the manuscript in the summary and discussion section.
- Hyperparameter tuning is critical in identifying an optimal neural network architecture. Authors should elaborate on this and show why particular learning rates, depths, and widths of NN are selected. The authors should show some results that indicate the selected parameters yield the optimal results with no overfitting observed. The results should be shown on test data which is never used for training the model.
We agree that hyperparameter tuning is an essential aspect of machine learning-based approaches. However, in the case of PINNs, there is no concern regarding overfitting, as it has been reported in [34]. In standard neural network training, labeled measured data are available, and the error function is constructed based on the deviation between the estimated and measured values. To achieve better generalization and performance on unseen test data, one should avoid overfitting to the training data. However, in PINN models used to solve PDEs, where the PDEs are imposed as constraints, it is not straightforward to clearly monitor the accuracy (training error) and generalization error (test error) because there is no access to the true solution.
Furthermore, in PINNs, the focus of hyperparameter tuning is to achieve a significant reduction in the total loss, which is composed of different loss terms. The objective is to find an optimal set of parameters that yield a substantial decrease in the objective value without significantly increasing the training time. Some works have explored the effect of hyperparameters on the accuracy of PINNs for certain PDE benchmarks [34,35]. In our study, we have chosen hyperparameters that have worked satisfactorily without significantly increasing the computational time, and some of our hyperparameter choices align with the benchmark studies in [34].
Therefore, we have added a discussion on hyperparameter tuning starting at line 268 in the revised manuscript.
- The literature survey on PINNs seems incomplete. The statement in line 77-79, "This previous studies..." is very general. Authors statement of PINNs for 3-D heat transfer is not convincing. Transient axis-symmetric heat transfer problems may be appropriate.
Thank you for the suggestion. We have included additional literature in the introduction section that is relevant to the use of PINNs in solving heat transfer problems.
Furthermore, we have adapted it throughout the manuscript to 2D transient heat transfer problem.
- It is recommended to include an illustration showing the various stages of the induction hardening process should be provided. All the boundary conditions and heat-loading directions should be shown.
Thank you for the suggestions. Some of the authors of the current manuscript have previously published an article that describes the various stages of the induction hardening process. For further details, interested readers can refer to the corresponding paper [29], which is cited in our manuscript.
We also appreciate the suggestion to improve Figure 1. We have made edits to make the boundaries and heat load directions clearer in the figure.
- The statement on line 58, "Artificial neural networks....invention in 1958..." is misleading as Frank Rosenblatt used a single feed-forward neuron "perceptron". Also, citations should be provided for various machine-learning tasks.
Thanks for the suggestion. The sentence is edited such that to mention the single perceptron in the original ANN.
We added the corresponding literatures to various machine learning tasks.
Round 2
Reviewer 1 Report
Accept.
Accept.